# ABCC1 and glutathione metabolism limit the efficacy of BCL-2 inhibitors in acute myeloid leukemia

Jessica Ebner [1,12], Johannes Schmoellerl [1,2,12], Martin Piontek[1], Gabriele Manhart[1], Selina Troester [1], Bing Z. Carter[3], Heidi Neubauer [4], Richard Moriggl [4], Gergely Szakács[5,6], Johannes Zuber [2,7], Thomas Köcher[8], Michael Andreeff [3], Wolfgang R. Sperr[9,10], Peter Valent [9,10] & Florian Grebien [1,11] ✉

The BCL-2 inhibitor Venetoclax is a promising agent for the treatment of acute myeloid leukemia (AML). However, many patients are refractory to Venetoclax, and resistance develops quickly. ATP-binding cassette (ABC) transporters mediate chemotherapy resistance but their role in modulating the activity of targeted small-molecule inhibitors is unclear. Using CRISPR/Cas9 screening, we find that loss of *ABCC1* strongly increases the sensitivity of AML cells to Venetoclax. Genetic and pharmacologic ABCC1 inactivation potentiates the anti-leukemic effects of BCL-2 inhibitors and efficiently re-sensitizes Venetoclax-resistant leukemia cells. Conversely, ABCC1 overexpression induces resistance to BCL-2 inhibitors by reducing intracellular drug levels, and high ABCC1 levels predicts poor response to Venetoclax therapy in patients. Consistent with ABCC1-specific export of glutathionylated substrates, inhibition of glutathione metabolism increases the potency of BCL-2 inhibitors. These results identify ABCC1 and glutathione metabolism as mechanisms limiting efficacy of BCL-2 inhibitors, which may pave the way to development of more effective therapies.

Acute myeloid leukemia (AML) is characterized by aberrant expansion of myeloid blast cells in the bone marrow and blood with subsequent suppression of normal hematopoiesis[1,2]. Mutations in transcription factors, epigenetic modifiers and key components of critical signaling pathways contribute to disease heterogeneity and disease evolution in AML[3,4]. The standard treatment of AML was based on chemotherapy and hematopoietic stem cell transplantation (HSCT) for several decades[5]. In the past years, several new treatment options have been developed in AML, such as demethylating agents and the use of targeted therapies[6,7]. Nevertheless, the prognosis of AML patients remains poor, as the efficacy of anti-leukemic therapies is often limited by their toxicity and by the development of resistance, resulting in disease relapse. Therefore, intense efforts have been made to identify novel targeted treatment options for AML patients.

[1]Institute for Medical Biochemistry, University of Veterinary Medicine Vienna, Vienna, Austria. [2]Research Institute of Molecular Pathology (IMP), Vienna BioCenter (VBC), Vienna, Austria. [3]Section of Molecular Hematology and Therapy, Department of Leukemia, The University of Texas MD Anderson Cancer Center, Houston, TX, USA. [4]Institute for Animal Breeding and Genetics, University of Veterinary Medicine Vienna, Vienna, Austria. [5]Center for Cancer Research, Medical University Vienna, Vienna, Austria. [6]Institute of Enzymology, Research Centre of Natural Sciences, Eötvös Loránd Research Network, Budapest, Hungary. [7]Medical University of Vienna, Vienna, Austria. [8]Vienna BioCenter Core Facilities, Vienna BioCenter, Vienna, Austria. [9]Department of Internal Medicine I, Division of Hematology and Hemostaseology, Medical University of Vienna, Vienna, Austria. [10]Ludwig Boltzmann Institute for Hematology and Oncology, Medical University of Vienna, Vienna, Austria. [11]St. Anna Children's Cancer Research Institute (CCRI), Vienna, Austria. [12]These authors contributed equally: Jessica Ebner, Johannes Schmoellerl. ✉e-mail: florian.grebien@vetmeduni.ac.at

Overexpression of anti-apoptotic proteins of the BCL-2 family, such as BCL-2, BCL-xL and MCL-1, is frequently found in various cancers, including AML, thereby perturbing homeostasis of survival pathways and protecting cancer cells from death[8]. Venetoclax is an oral, selective BCL-2 inhibitor that mimics the binding of pro-apoptotic BH3-only proteins to BCL-2, thereby effectively inducing apoptosis[9,10]. The FDA has approved Venetoclax in combination with either low-dose Cytarabine (LDAC), Azacytidine or Decitabine for the treatment of newly diagnosed AML patients who do not qualify for intensive induction chemotherapy or are over 75 years old[11]. Venetoclax is currently being evaluated in more than 300 clinical trials (https://clinicaltrials.gov/), which further highlights it as a promising novel treatment option in cancer.

Despite promising results in AML therapy, up to one third of patients is refractory to Venetoclax-based therapy, and resistance often develops within one year of treatment[12], necessitating studies of cellular mechanisms that modulate Venetoclax responsiveness. For example, upregulation of the anti-apoptotic protein MCL-1 in leukemia cells can lead to Venetoclax resistance[13]. Reduced mitochondrial priming, by displacing pro-apoptotic proteins from drug-targeted BCL-2 to non-targeted MCL-1 also induces resistance to BCL-2 antagonists in AML cells[14]. In accordance, MCL-1 inhibitors have shown pre-clinical activity in AML[15,16], and MCL-1 inhibition synergized with Venetoclax in AML models[17–19]. Recently, genome-wide CRISPR/Cas9-mediated loss-of-function screens showed that loss of TP53, BAX or PMAIP1 renders AML cells resistant to Venetoclax[20]. The combination of Venetoclax with an MDM2 inhibitor was highly synergistic in AML PDX models[21]. In addition, targeting the mitochondrial protein CLPB sensitized cells to Venetoclax treatment due to cristae remodeling and degradation of targets in the respiratory chain[22–24]. While mechanisms of resistance to Venetoclax that directly affect BCL-2 have been explored, changes influencing the cellular pharmacokinetics of BCL-2 inhibitors and their accumulation in leukemic target cells are largely unknown.

ATP-binding cassette (ABC) transporters are important regulators of drug efflux. By transporting substrates across the cell membrane, they play critical roles for multiple physiological processes, including detoxification, metabolism and cell signaling. Substrates of ABC transporters cover a broad spectrum of targets including lipids, hormones, ions, nucleosides, metabolites and xenobiotics, but also cytotoxic drugs[25].

The 48 ABC transporters found in humans have been classified into seven subgroups (ABCA-ABCG) that differ in structure, function and tissue expression[26]. In line with the potential of ABC transporters to efflux anticancer agents, their overexpression has been associated with poor response to chemotherapy and multidrug resistance (MDR)[27]. While ABCB1, ABCC1 and ABCG2 have been most prominently studied in the context of MDR, at least 16 other ABC transporters have the potential to efflux anticancer drugs[28].

High ABCC1 expression was associated with poor disease-free survival[29], and co-expression of multiple ABC transporters worsened the prognosis of AML patients[30]. Thus, while the role of ABC transporters in mediating resistance to chemotherapeutic agents is well established, it is unclear if and how they modulate the response of cancer cells to targeted therapies.

Here, we used a targeted CRISPR/Cas9-mediated loss-of-function screening approach to systematically interrogate the role of ABC transporters in modulating the response of AML cells to Venetoclax treatment. Genetic or pharmacological inactivation of ABCC1 potentiated the anti-leukemic effects of BH3 mimetic drugs. Inactivation of ABCC1 was sufficient to re-sensitize resistant leukemia cells to Venetoclax treatment and high ABCC1 expression predicted inferior responses to Venetoclax in AML patients. Finally, we uncovered a role for glutathione metabolism in modulating the efficiency of BCL-2 inhibitors. Thus, our work highlights new entry points for improving targeted therapy using BH3 mimetics in AML and beyond.

## Results

### ABCC1 modulates Venetoclax sensitivity in AML cells

To investigate the essentiality of the 48 human ABC-transporters in drug-naive AML cells, we tested how loss of each of them affects AML cell growth using a CRISPR/Cas9-based approach. The effects of gene perturbations were measured in competitive proliferation assays of Cas9- and sgRNA/IRFP670-expressing MOLM-13 AML cells over time (Supplementary Fig. 1A). If mutational disruption of a gene negatively affects cell proliferation/viability, the levels of IRFP670-positive cells decrease, as exemplified by knockout of the essential gene encoding the 60 S ribosomal protein L17 (RPL17). In contrast, targeting the adeno-associated virus integration site 1 locus (AAVS1) did not affect the levels of IRFP670-positive cells and hence served as negative control. While 44 out of 48 ABC transporters were not essential for the growth of AML cells, targeting of ABCF1, ABCA3, TAP2 and ABCE1 led to decreased cellular fitness (Supplementary Fig. 1A). These transporters were highly abundant in AML cell lines whereas the majority of other ABC transporters was expressed at moderate or low levels (Supplementary Fig. 1B), which is in line with the available literature[29,31–34].

Next, we used the same CRISPR/Cas9-mediated gene silencing strategy to assess the role of the 44 nonessential human ABC transporters (Supplementary Fig. 1A) in the response of MOLM-13 cells to Venetoclax or the chemotherapeutic Cytarabine. Upon establishment of balanced ratios between sgRNA-transduced (IRFP670-positive) and non-transduced cells, cultures were treated with low concentrations of Venetoclax that did not interfere with the proliferation of MOLM-13 cells (1 nM; $GI_{10}$) or DMSO. Ratios of IRFP670-positive to IRFP670-negative cells were monitored over time (Fig. 1A). By comparing the area under the curve (AUC) of IRFP670-positive cells over time upon drug treatment versus DMSO controls we identified synergistic and/or antagonistic interactions between genetic inactivation of ABC transporters and Venetoclax exposure (Supplementary Fig. 1C). While no antagonistic interactions were observed, this screening approach identified synergistic effects between loss of ABCC1 or ABCA2 and Venetoclax treatment (Fig. 1B, left). However, the same screening scheme did not reveal synergistic or antagonistic effects when mutational disruption of ABC transporters was combined with Cytarabine (Fig. 1B, right). The interaction between ABCA2 knockout and Venetoclax treatment could not be validated in independent experiments (Supplementary Fig. 1D). In contrast, CRISPR/Cas9-mediated disruption of ABCC1 potentiated Venetoclax sensitivity of MOLM-13 and HL-60 cells (Fig. 1C). This strong, synergistic effect was also observed in murine MLL-AF9/NrasG12D-driven leukemia cells (RN2)[35] (Supplementary Fig. 1E).

ABCC1, also known as multidrug resistance protein 1 (MRP1), is a member of the Multidrug Resistance Associated Protein (MRP) subfamily[36] and transports organic cations and conjugated substrates, including oxidized glutathione (glutathione disulfide, GSSG)[37,38].

To validate the interaction between ABCC1 and Venetoclax in more detail, we established single-cell-derived MOLM-13 clones expressing ABCC1- or AAVS1-targeting sgRNAs. Clonal cell lines expressing sgABCC1 harbored nonsense mutations on both alleles of the ABCC1 gene (Supplementary Fig. 1F). While all clonal cell lines proliferated at the same rate in the presence of DMSO, growth of ABCC1 knockout clones was selectively impaired upon Venetoclax treatment (Supplementary Fig. 1G).

Taken together, these results identify the ABCC1 gene as a critical modulator of Venetoclax sensitivity in AML cells.

### Pharmacologic inhibition of ABCC1 synergizes with BCL-2-inhibitors

We next aimed to assess the potential synergy between combined targeting of ABCC1 and BCL-2 using a pharmacological approach. In line with genetic data, AML cell lines were resistant to ABCC1 inhibition by the pyrazolopyrimidine Reversan[39] or the leukotriene D4 receptor

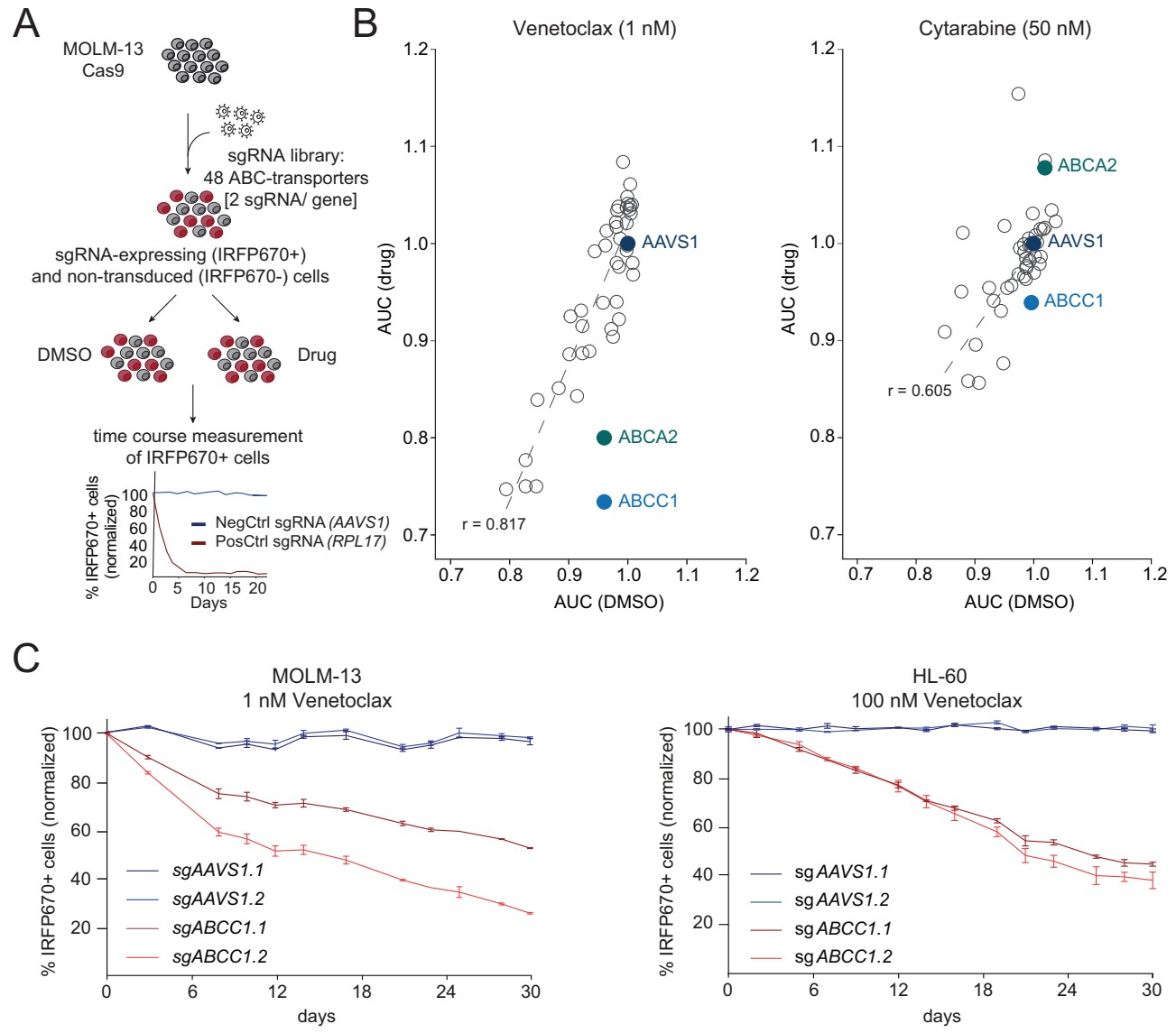

**Fig. 1 | *ABCC1* modulates Venetoclax sensitivity in AML cells. A** Schematic representation of the CRISPR/Cas9-based competitive proliferation assay. Each ABC transporter gene is targeted with two sgRNAs in Cas9-expressing MOLM-13 cells. Cells were divided and treated with DMSO or Venetoclax. The percentage of sgRNA-expressing cells (IRFP670⁺) was monitored over time and the area under the curve (AUC) was determined for each sgRNA. The corresponding flow cytometric gating strategy for IRFP670⁺ cells is depicted in Supplementary Fig. 6A. **B** The AUC after 30 days of treatment of DMSO-treated MOLM-13-Cas9 cells expressing sgRNAs targeting ABC transporters is plotted against the AUC (30 days) of cells treated with 1 nM Venetoclax (left) or 50 nM Cytarabine (right). *n* = 4 experimental replicates with 2 different sgRNAs per gene. **C** Competitive proliferation assay of MOLM-13-Cas9 cells treated with 1 nM Venetoclax (left; *n* = 2 experimental replicates) and HL-60-Cas9 cells treated with 100 nM Venetoclax (right; *n* = 3 experimental replicates) for 30 days. Percentages of sgRNA/IRFP670+ cells were normalized to day 0 of treatment and to DMSO controls. Data are presented as mean values ± SD. **B**, **C** Source data are provided as a Source Data file.

antagonist MK-571[40] (Supplementary Fig. 2A). Strikingly, Reversan-mediated ABCC1 inhibition indeed also exhibited strong synergistic effects with Venetoclax in MOLM-13 cells (Fig. 2A). Addition of a single dose of 5 μM Reversan 24 h prior to Venetoclax administration drastically sensitized MV4-11 and HL-60 cells to BCL-2 inhibition, resulting in a 3.9 and 6.5-fold decrease of the GI$_{50}$, respectively (Fig. 2B). To explore the interaction between BCL-2-targeting small molecules and ABCC1 in more detail, we included other BH3-mimetic drugs that inhibit various BCL-2-family members (Fig. 2C). We found strong synergistic effects between ABCC1 inhibition and the BH3-mimetics Venetoclax, Navitoclax (BCL-2/BCL-xL inhibitor) and ABT-737 (BCL-2/BCL-xL/BCL-w inhibitor) in different AML cell lines (Fig. 2D and Supplementary Fig. 2B–E). In contrast, no synergy was found between Reversan-mediated ABCC1 inhibition and Cytarabine or other targeted therapies, including the tyrosine kinase inhibitors Midostaurin and

Gilteritinib, the IDH1 inhibitor Ivosidenib and the IDH2 inhibitor Enasidenib (Fig. 2D and Supplementary Fig. 2C–E).

Taken together, these data show that pharmacologic inhibition of ABCC1 sensitizes AML cells towards BH3-mimetic drugs targeting BCL-2/BCL-xL/BCL-w.

## The dual BCL-2/BCL-xL inhibitor AZD-4320 strongly synergizes with ABCC1-inhibition in vitro and in vivo

AZD-4320 was recently presented as a novel BH3 mimetic drug that inhibits both BCL-2 and BCL-xL (Fig. 3A). Compared to BCL-2 inhibition, AZD-4320 showed improved activity in patient-derived AML cells and enhanced in vivo treatment efficacy in xenograft models[41]. We therefore aimed to investigate if the activity of AZD-4320 could also be enhanced by targeting *ABCC1*. In competitive proliferation assays in MOLM-13 and HL-60 cells, treatment with AZD-4320 resulted in strong

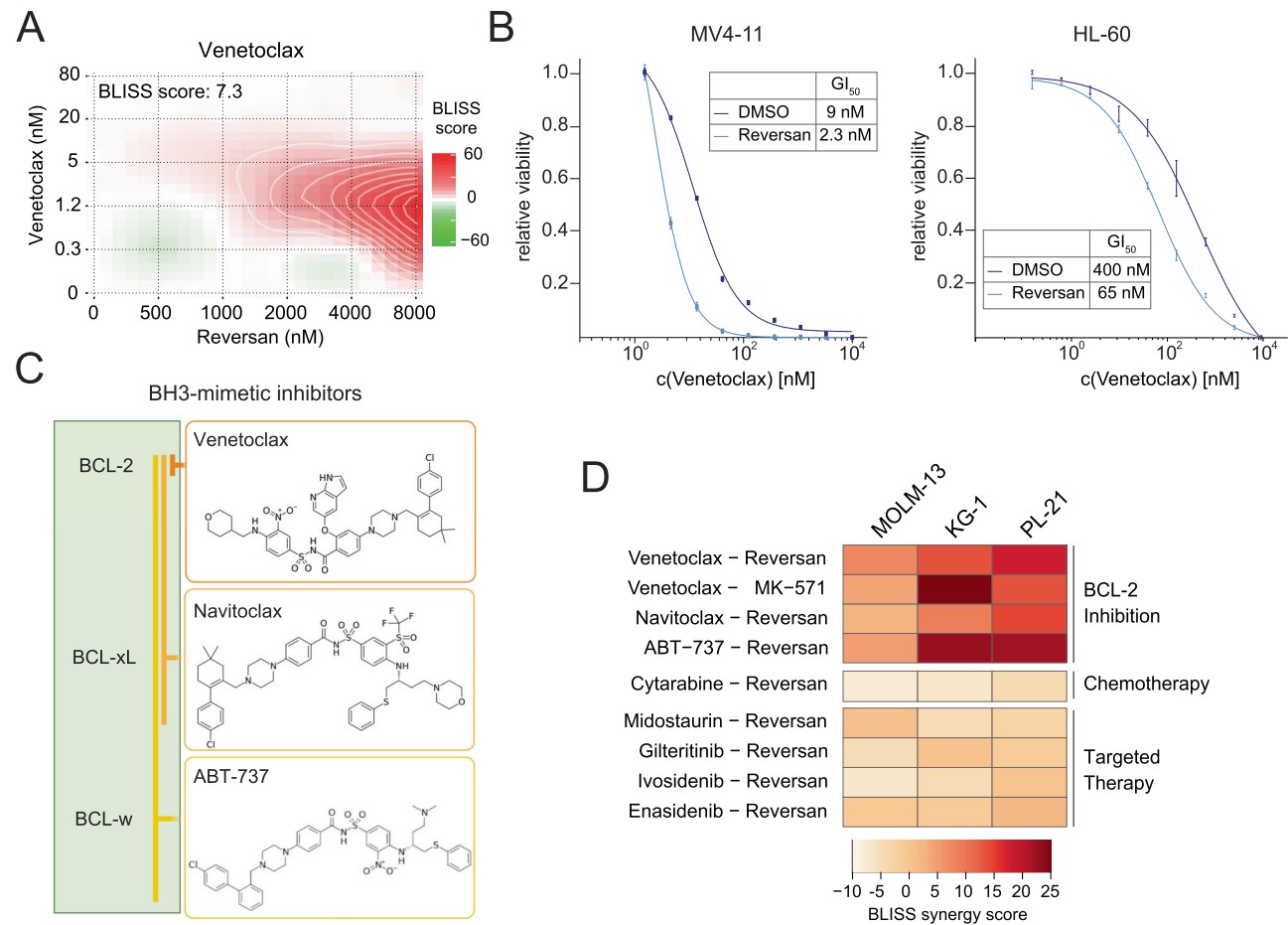

**Fig. 2 | Pharmacologic inhibition of ABCC1 synergizes with BCL-2-inhibitors.**
**A** BLISS synergy score distribution of MOLM-13 cells treated with Venetoclax at indicated concentrations in combination with Reversan for 5 days, determined using SynergyFinder. Data are represented as the relative deviation from BLISS additivity. $n = 2$ experimental replicates. **B** Five-day dose–response curves of Venetoclax in MV4-11 (left) and HL-60 (right) cells, co-treated with either DMSO or 5 μM Reversan. Data are presented as mean values ± SD. $n = 3$ experimental replicates. **C** Structure of BH3-mimetics inhibiting the anti-apoptotic proteins BCL-2, BCL-xL and BCL-w. **D** Heatmap depicting the overall relative deviation from BLISS additivity (BLISS synergy score), determined by SynergyFinder of three different AML cell lines (MOLM-13, KG-1, PL-21) treated with Reversan or MK-571 with indicated drugs for five days. $n = 2$ experimental replicates. **A**, **B**, **D** Source data are provided as a Source Data file.

depletion of cells expressing *ABCC1*-targeting sgRNAs (Fig. 3B) while pharmacological ABCC1-inhibition using Reversan exhibited a synergistic growth inhibitory effect together with AZD-4320 in HL-60 cells (Supplementary Fig. 3A). Consistent with reports showing that the topoisomerase II-inhibitor Etoposide is a substrate of ABCC1[42], the GI$_{50}$ of MOLM-13 *ABCC1*-knockout clones to Etoposide was 1.4– 2.4-fold lower than in control cells (Supplementary Fig. 3B). Ectopic re-expression of ABCC1 in *ABCC1*-knockout cells reverted this effect (Supplementary Fig. 3C). Knockout of *ABCC1* led to a 1.9 - 2.8-fold reduction of the GI$_{50}$ concentrations of Venetoclax and AZD-4320, while no difference in Midostaurin sensitivity was observed upon inactivation of *ABCC1* (Supplementary Fig. 3B). In line, cell numbers of MOLM-13 *ABCC1*-knockout clones were massively reduced by exposure to AZD-4320 in concentrations that did not affect control cells (Fig. 3C).

To evaluate if knockout of *ABCC1* also enhances the anti-leukemic effects of AZD-4320 in vivo, we transplanted equal numbers of MOLM-13 *AAVS1.1* control or *ABCC1*-knockout cells into NSG-S mice and treated them once a week with either AZD-4320 (10 mg/kg) or vehicle. While the engraftment rates were similar between cohorts, the leukemia burden was significantly reduced in mice transplanted with MOLM-13 *ABCC1*-knockout cells upon AZD-4320 treatment compared to vehicle treatment and to MOLM-13 *AAVS1.1* control cells (Fig. 3D, E). Accordingly, recipients of MOLM-13 *ABCC1*-knockout cells harbored

significantly less AML cells in their bone marrow upon AZD-4320 treatment than mice transplanted with MOLM-13 *AAVS1.1* control cells (Fig. 3F).

Together, these data show that loss of *ABCC1* increases the sensitivity of leukemia cells to AZD-4320 treatment in vivo and in vitro.

### Overexpression of ABCC1 promotes resistance to BCL-2 inhibitors by reducing intracellular drug concentrations

ABCC1 overexpression has been implicated in chemotherapy resistance and is associated with adverse prognosis in a variety of malignant diseases[43,44]. To investigate whether ABCC1 overexpression can drive resistance to BCL-2 inhibition, we used HL-60 cells that were engineered to ectopically express ABCC1 (Fig. 4A, B). ABCC1-overexpressing cells were resistant to Etoposide and the BH3 mimetics Venetoclax and AZD-4320, but not to Midostaurin (Fig. 4C). Importantly, ectopic ABCC1 re-expression in MOLM-13 *ABCC1*-knockout cells reversed the AZD-4320 sensitivity of *ABCC1*-deficient cells and induced resistance to this drug (Fig. 4D, E). Taken together, our results strongly suggested that ABCC1 expression protects AML cells by mediating the efflux of BH3 mimetics.

To investigate if ABCC1 is involved in efflux of Venetoclax or AZD-4320 we treated HL-60 cells overexpressing ABCC1 with either compound and used liquid-chromatography mass-spectrometry (LC-MS) to quantify drug concentrations in cell lysates and fresh culture

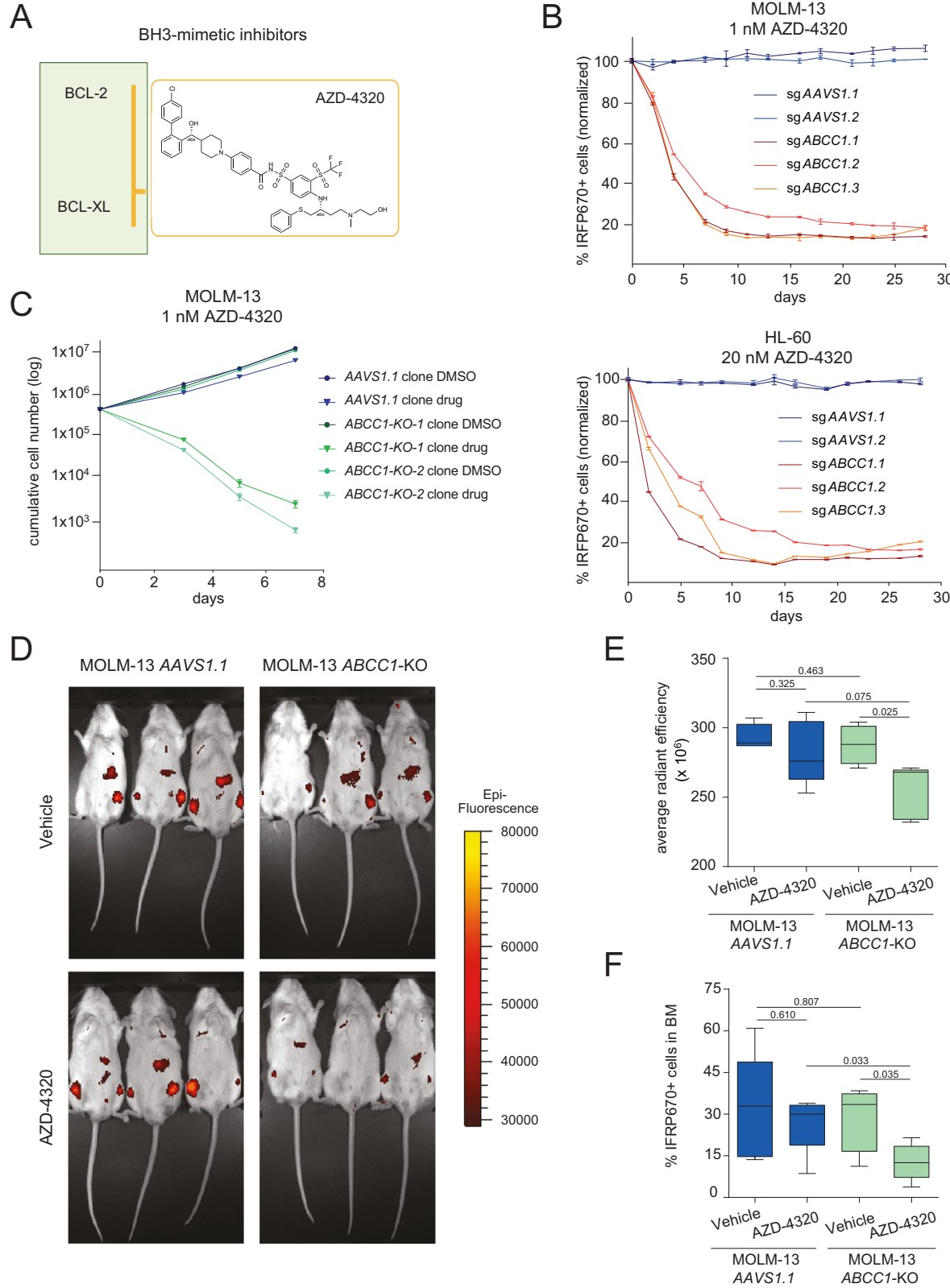

supernatants over time (Supplementary Fig. 3D, E). While the amounts of Venetoclax and AZD-4320 decreased in cell lysates over time, we observed an increase in drug concentrations in culture supernatants. Moreover, the intracellular drug concentrations were lower in ABCC1-overexpressing HL-60 than in wild-type HL-60 cells upon treatment (Fig. 4F), and Reversan-mediated ABCC1 inhibition resulted in an accumulation of intracellular Venetoclax (Supplementary Fig. 3F).

The fluorescent cell viability marker Calcein-acetoxymethyl ester (Calcein-AM) is a substrate of ABCC1 and can be used as a sensitive fluorometric reporter for ABC transporter function[45,46]. Accordingly, ABCC1 overexpression caused reduced intracellular Calcein-AM levels, while ABCC1 inhibition by Reversan increased them (Fig. 4G). We reasoned that if BCL-2 inhibitors are ABCC1 substrates, they would act as competitive inhibitors of Calcein AM export. Indeed, treatment of

**Fig. 3 | The dual BCL-2/BCL-xL inhibitor AZD-4320 strongly synergizes with ABCC1-inhibition in vitro and in vivo. A** Structure of the dual BCL-2/BCL-xL inhibitor AZD-4320. **B** Competitive proliferation assay of MOLM-13-Cas9 cells treated with 1 nM AZD-4320 (top) and HL-60-Cas9 cells treated with 20 nM AZD-4320 (bottom) for 29 days. Percentages of sgRNA/IRFP670+ cells were normalized to day 0 of treatment and to DMSO controls. Data are presented as mean values ± SD. $n = 3$ experimental replicates. **C** Growth curves of MOLM-13-Cas9 knockout clones (*AAVS1.1, ABCC1-KO-1, ABCC1-KO-2*) treated with DMSO or 1 nM AZD-4320. Data are presented as mean values ± SD. $n = 2$ experimental replicates. **D** Bioluminescence imaging of vehicle- or AZD-4320 treated mice [10 mg/kg] 16 days after transplantation with clonal MOLM-13-Cas9 *AAVS1.1* or *ABCC1-KO-1* cells. **E** Signal quantification of the epi-fluorescence in vivo imaging (IRFP670).

Boxes represent interquartile ranges, horizontal lines represent the mean, whiskers indicate lower and upper limits. Significance was determined with an unpaired Student's $t$ test with two-tailed $P$ value as indicated; $n = 4$ (MOLM-13 *ABCC1-KO-1* vehicle)/$n = 5$ (MOLM-13 *ABCC1-KO-1* treatment, MOLM-13 *AAVS1.1* vehicle and treatment) mice. **F** Quantification of the percentage of sgRNA/IRFP670+ cells in the bone marrow of mice treated with vehicle or AZD-4320 [10 mg/kg] 16 days after transplantation with clonal MOLM-13-Cas9 *AAVS1.1* or *ABCC1-KO-1* cells. Boxes represent interquartile ranges, horizontal lines represent the mean, whiskers indicate lower and upper limits. Significance was determined with an unpaired Student's $t$ test with two-tailed $P$ value as indicated; $n = 4$ mice. The corresponding flow cytometric gating strategy for IRFP670$^+$ cells is depicted in Supplementary Fig. 6B. **B, C, E, F** Source data are provided as a Source Data file.

HL-60 cells with increasing concentrations of Venetoclax or AZD-4320 caused a dose-dependent intracellular accumulation of Calcein-AM (Fig. 4H, left). The same effect was also observed in ABCC1-overexpressing HL-60 cells, but only at much higher drug concentrations (Fig. 4H, right), suggesting that the levels of ABCC1 expression tune the intracellular efflux of Calcein-AM and BH3 mimetics, respectively.

In summary, these data show that BH3 mimetics are substrates of ABCC1 and that overexpression of ABCC1 limits the sensitivity of AML cells to BCL-2 inhibitors by reducing their intracellular concentration.

## ABCC1 inhibition reverses Venetoclax-resistance in AML cells

Resistance to Venetoclax represents a significant clinical problem. Therefore, we assessed whether ABCC1's role in modulating intracellular concentrations of BCL-2 inhibitors could be exploited to target Venetoclax-resistant AML. The p53-deficient human AML cell line THP-1 tolerates high concentrations of Venetoclax (GI$_{50}$ 5 μM) (Fig. 5A). CRISPR/Cas9-mediated targeting of *ABCC1* in Cas9-expressing THP-1 cells using multiple sgRNAs re-sensitized them to BCL-2 inhibition, as *ABCC1*-targeted cells were rapidly depleted upon Venetoclax treatment (Fig. 5B) In accordance, Reversan-mediated ABCC1 inhibition together with Venetoclax treatment impaired the growth of THP-1 cells (Fig. 5C). CRISPR/Cas9-mediated targeting of *ABCC1* in THP-1 cells significantly increased intracellular Venetoclax levels as compared to THP-1 *AAVS1.1* control cells (Fig. 5D).

To investigate the involvement of ABCC1 in mechanisms of Venetoclax resistance in more detail, we treated Cas9-expressing MOLM-13 and MV4-11 AML cells with increasing concentrations of Venetoclax until they displayed stable growth in culture in the presence of 1 μM Venetoclax (Fig. 5E). Venetoclax-resistant and -sensitive cells did not differ in *ABCC1* mRNA expression (Supplementary Fig. 3G), indicating that induction of Venetoclax resistance did not result in ABCC1 upregulation. Interestingly, Venetoclax-resistant MOLM-13 cells were more sensitive to the MCL-1 inhibitor AZD-5991 compared to parental cells, while being highly resistant to Venetoclax and AZD-4320 (Supplementary Fig. 3H). Yet, CRISPR/Cas9-mediated targeting of *ABCC1* restored Venetoclax sensitivity in resistant MOLM-13 cells (Supplementary Fig. 3I). In addition, Reversan-mediated ABCC1 inhibition in Venetoclax-resistant MV4-11 cells restored their Venetoclax sensitivity, resulting in GI$_{50}$ concentrations similar to MV4-11 wild-type cells (Fig. 5F). Accordingly, combined Reversan and Venetoclax treatment led to a significant growth inhibition of Venetoclax-resistant MV4-11 cells (Fig. 5G).

Taken together, these data show that although ABCC1 did not contribute to the acquired resistance to Venetoclax, inhibition of ABCC1 can restore Venetoclax sensitivity in resistant AML cells.

## ABCC1 levels dictate the response of AML to Venetoclax

Expression analysis of ABC transporters in AML patients in the BeatAML dataset[47] showed that unlike AML cell lines (Supplementary Fig. 1B), many of the 48 human ABC transporters were expressed at different levels in Venetoclax-naive patients (Supplementary Fig. 4A).

*ABCC1* was expressed at highest levels, followed by *ABCA2* and *ABCA7*. *ABCC1* was also overexpressed in AML cells compared to normal peripheral blood mononuclear cells (PBMCs) or bone marrow of healthy donors (Fig. 6A, B). Furthermore, *ABCC1* was particularly highly expressed in AML subtypes FAB M0, M1 and M2 (Supplementary Fig. 4B). To investigate if expression of ABC transporters correlates with response to Venetoclax treatment in AML patients, we first quantified the expression of several ABC transporters that are relevant in AML in a cohort of samples from 16 patients prior to initiation of Venetoclax treatment. Also in these patients, *ABCC1* was the ABC transporter with the highest expression levels. *ABCB1, ABCC3, ABCC4* and *ABCC5* were expressed at intermediate levels while ABCC10 was expressed at low levels. In contrast, expression of *ABCG2, ABCC2* and *ABCC6* was not detected (Fig. 6C and Supplementary Table 1). We then grouped patients into two categories based on their response to Venetoclax therapy. While good responders ($n = 4$) achieved complete remission (CR) or a complete remission with incomplete hematologic recovery (CRI), poor responders ($n = 10$) only reached cytoreduction or stable disease after Venetoclax treatment (Supplementary Table 1). This analysis revealed that poor responders expressed significantly higher levels of *ABCC1* compared to good responders (Fig. 6D). In contrast, no significant difference in the expression of all other analyzed ABC transporters was found between these patient groups (Supplementary Fig. 4C). All but two patients had been pre-treated with 5-Azacytidine or chemotherapy (Supplementary Table 1). Secondary AML (sAML) is associated with frequent Venetoclax resistance[48]. In our cohort sAML patients exhibited slightly higher ABCC1 expression levels compared to patients with de novo AML (Supplementary Fig. 4D).

We next aimed to investigate if pharmacologic inhibition of ABCC1 could improve the potency of Venetoclax in primary patient-derived AML cells. Leukemia cells were more sensitive to the combination of Venetoclax and Reversan-mediated ABCC1 inhibition than to Venetoclax alone (Supplementary Fig. 4E), and this synergistic effect was even stronger with the AZD-4320-Reversan combination, inducing a 5- to 20-fold reduction in GI$_{50}$ values in all tested samples (Fig. 6E).

In summary, *ABCC1* is highly expressed in primary AML cells. Moreover, our data suggest that high *ABCC1* expression predicts poor response to Venetoclax therapy. Finally, we show that pharmacologic inhibition of ABCC1 potentiates the anti-leukemic effects of BCL-2 inhibitors in primary AML cells.

## Glutathione metabolism modulates the sensitivity to BH3 mimetics

Glutathione S-transferases (GST) are detoxifying enzymes that protect mammalian macromolecules from reactive electrophilic substances using glutathione (GSH) as a xenobiotic acceptor[49]. In contrast to most other ABC transporters[50], ABCC1 is able to efflux drugs in the presence of GSH and/or in the form of GSH-conjugates[51]. The rate-limiting step in the glutathione synthesis pathway is catalyzed by glutamate cysteine ligase (GCL), which can be inhibited using Buthionine Sulfoximine (BSO)[52] (Fig. 7A). GSTs, in turn, conjugate GSH to xenobiotics

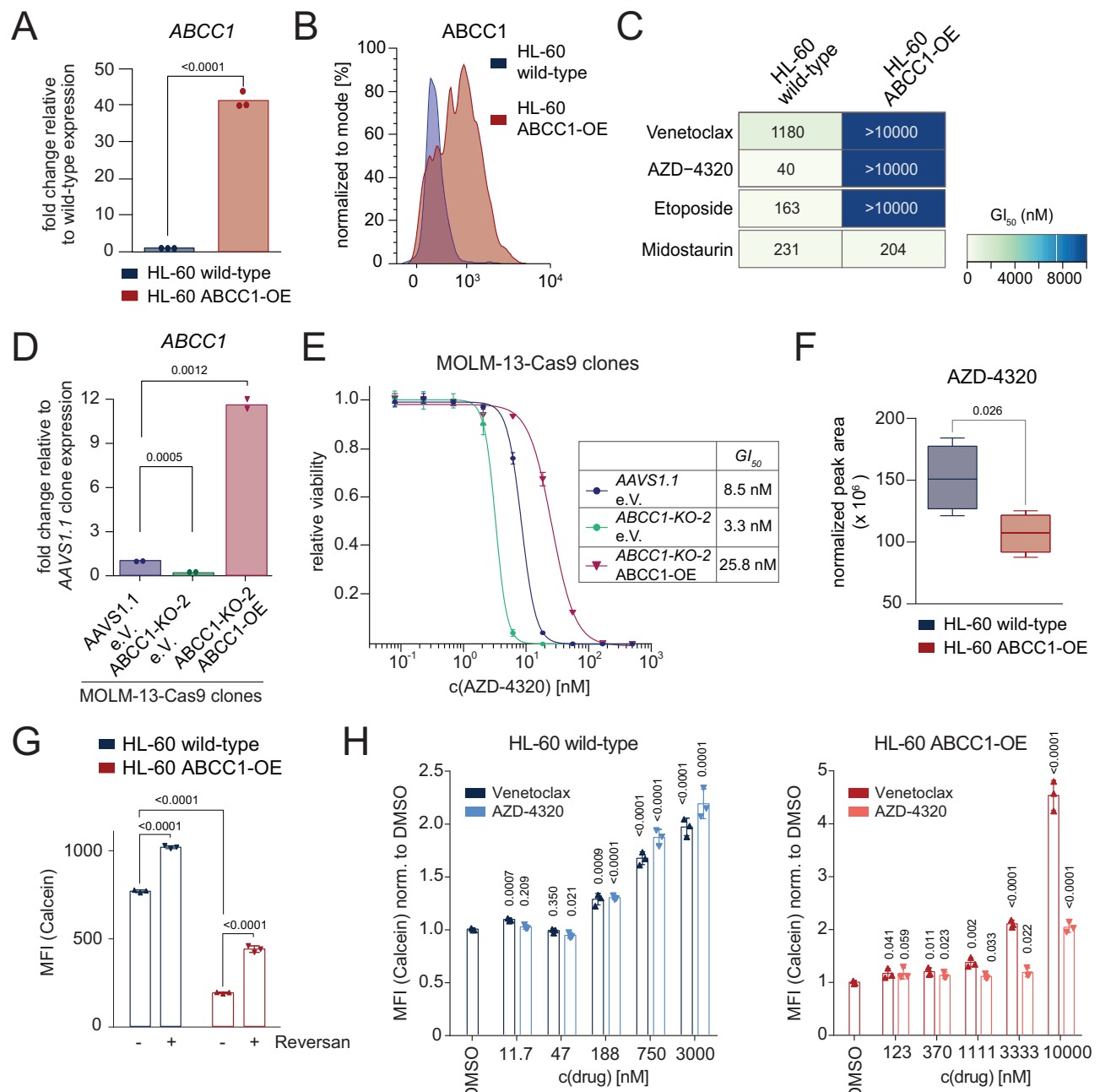

**Fig. 4 | Overexpression of ABCC1 promotes resistance to BCL-2 inhibitors by reducing intracellular drug concentrations. A** RT-qPCR analysis of *ABCC1* mRNA expression in HL-60 wild-type (WT) and ABCC1-overexpressing (OE) cells. Fold change of *ABCC1* expression in ABCC1-OE cells compared to WT. Data are presented as mean values. Significance was determined with an unpaired Student's *t* test with two-tailed *P* value as indicated. *n* = 3 technical replicates, *n* = 1 experimental replicate. **B** Representative flow cytometric measurement of intracellular ABCC1 levels in HL-60 WT and ABCC1-OE cells. The corresponding gating strategy is depicted in Supplementary Fig. 6C. **C** Heatmap of GI$_{50}$ values [nM] of 5-day viability assays with indicated drugs in HL-60 WT and ABCC1-OE cells. *n* = 3 experimental replicates. **D** RT-qPCR analysis of ABCC1 mRNA expression of MOLM-13-Cas9 *AAVS1.1* and *ABCC1-KO-2* clones, transduced with either empty vector (e.V.) or an ABCC1-overexpressing (OE) construct. Levels were normalized to ACTB levels and to the expression of MOLM-13-Cas9 *AAVS1.1* e.V. Data are presented as mean values. Significance was determined with an unpaired Student's *t* test with two-tailed *P* value as indicated. *n* = 2 technical replicates, *n* = 1 experimental replicate. **E** Dose−response curves of AZD-4320 in MOLM-13-Cas9 *AAVS1.1* or *ABCC1-KO-2 clones* transduced with either empty vector (e.V.) or an ABCC1-OE construct after 5 days of treatment. Data are presented as mean values ± SD. *n* = 3 experimental

replicates. **F** Intracellular concentrations of AZD-4320 in HL-60 WT and ABCC1-OE cells determined using LC-MS/MS. The normalized peak area represents the relative compound concentration in the samples. Combined results of hydrophilic interaction liquid chromatography (HILIC) and reversed phase chromatography (RP) are depicted. Boxes represent interquartile ranges, horizontal lines represent the mean, whiskers indicate lower and upper limits. Significance was determined with an unpaired Student's *t* test with two-tailed *P* value as indicated. *n* = 2 experimental replicates. **G** Median fluorescence intensity (MFI) of Calcein in HL-60 WT and ABCC1-OE cells treated either with DMSO or Reversan [10 µM] for 45 h. The corresponding flow cytometric gating strategy is depicted in Supplementary Fig. 6D. Data are presented as mean values ± SD. Significance was determined with an unpaired Student's *t* test with two-tailed *P* value as indicated. *n* = 3 experimental replicates. **H** Median fluorescence intensity (MFI) of Calcein normalized to DMSO in HL-60 WT and ABCC1-OE cells treated with DMSO, Venetoclax or AZD-4320 at indicated concentrations for 45 h. Data are presented as mean values ± SD. Significance compared to DMSO was determined with an unpaired Student's *t* test with two-tailed *P* value as indicated. *n* = 3 experimental replicates. The corresponding flow cytometric gating strategy is depicted in Supplementary Fig. 6D. **A**, **C**−**H** Source data are provided as a Source Data file.

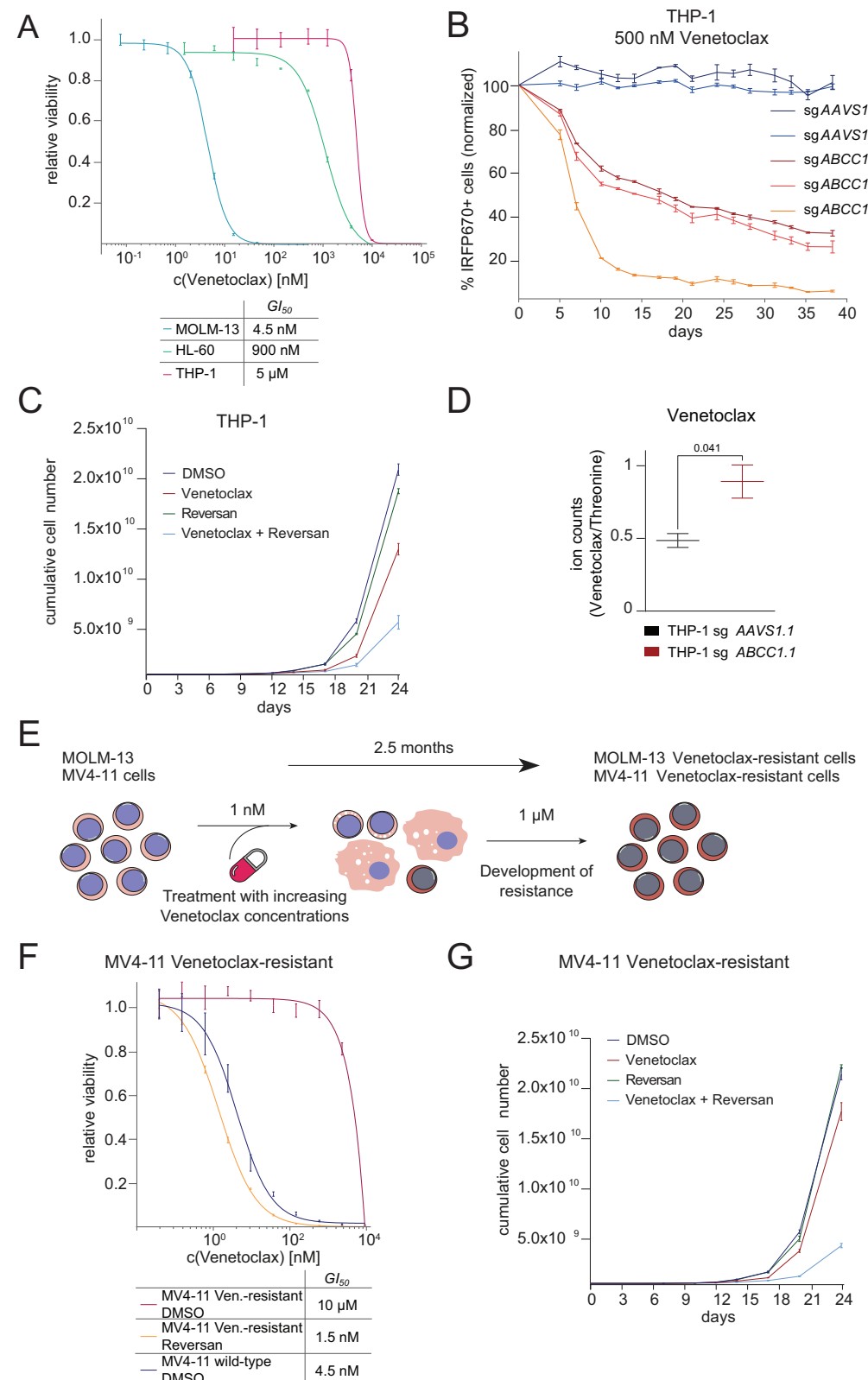

in a first step of detoxification[53,54]. The diuretic ethacrynic acid (EA) is a potent and reversible inhibitor of GSTs[55]. Thus, we aimed to test whether perturbation of GSH synthesis and -conjugation modulates the response of leukemia cells to BH3-mimetics. While BSO alone did not affect cell viability, BSO pre-treatment strongly increased Venetoclax sensitivity of HL-60 cells (Supplementary Fig. 5A, B). Moreover, ABCC1-overexpressing HL-60 cells were efficiently re-sensitized to

Venetoclax through pre-treatment with BSO (Supplementary Fig. 5B). Combinatorial treatment with Venetoclax or AZD-4320 and BSO severely impaired the growth of MOLM-13 cells (Fig. 7B and Supplementary Fig. 5C), and BSO treatment re-sensitized Venetoclax-resistant MOLM-13 cells to BCL-2 inhibition (Fig. 7C). The combinatorial effect of BSO and BCL-2 inhibitors was stronger in MOLM-13 *ABCC1*-knockout cells than in control clones (Fig. 7D and Supplementary Fig. 5D). These

**Fig. 5 | ABCC1 inhibition reverses Venetoclax-resistance in AML cells.**
**A** Dose−response curves of Venetoclax in MOLM-13, HL-60 and THP-1 cells after 5 days of treatment. Data are presented as mean values ± SD. $n = 3$ experimental replicates. **B** Competitive proliferation assay of THP-1-Cas9 cells treated with 500 nM Venetoclax for 38 days. Percentages of sgRNA/IRFP670+ cells were normalized to day 0 of treatment and to DMSO controls. Data are presented as mean values ± SD. $n = 3$ experimental replicates. The corresponding flow cytometric gating strategy for IRFP670+ cells is depicted in Supplementary Fig. 6A. **C** Growth curves of THP-1 cells treated with DMSO, 1 µM Venetoclax, 2 µM Reversan or 1 µM Venetoclax in combination with 2 µM Reversan. Data are presented as mean values ± SD. $n = 3$ experimental replicates. **D** Ion count ratio of intracellular Venetoclax to Threonine (control amino acid) as determined by LC-MS/MS of Cas9-

expressing THP-1 cells transduced with either sg*AAVS1.1* or sg*ABCC1.1*. Data are presented as mean values ± SD. $n = 2$ experimental replicates. Significance was determined with an unpaired Student's *t* test with two-tailed *P* value as indicated. **E** Schematic representation of the generation of Venetoclax-resistant MOLM-13 and MV4-11 cells. Cells were treated up to 2.5 months using increasing concentrations of Venetoclax, until they showed stable growth in media supplemented with 1 µM Venetoclax. **F** Dose−response curves of Venetoclax in MV4-11 wild-type and Venetoclax-resistant cells co-treated with either DMSO or 5 µM Reversan for 5 days. Data are presented as mean values ± SD. $n = 3$ experimental replicates. **G** Growth curves of Venetoclax-resistant MV4-11 treated with DMSO, 1 µM Venetoclax, 2 µM Reversan or 1 µM Venetoclax in combination with 2 µM Reversan. Data are

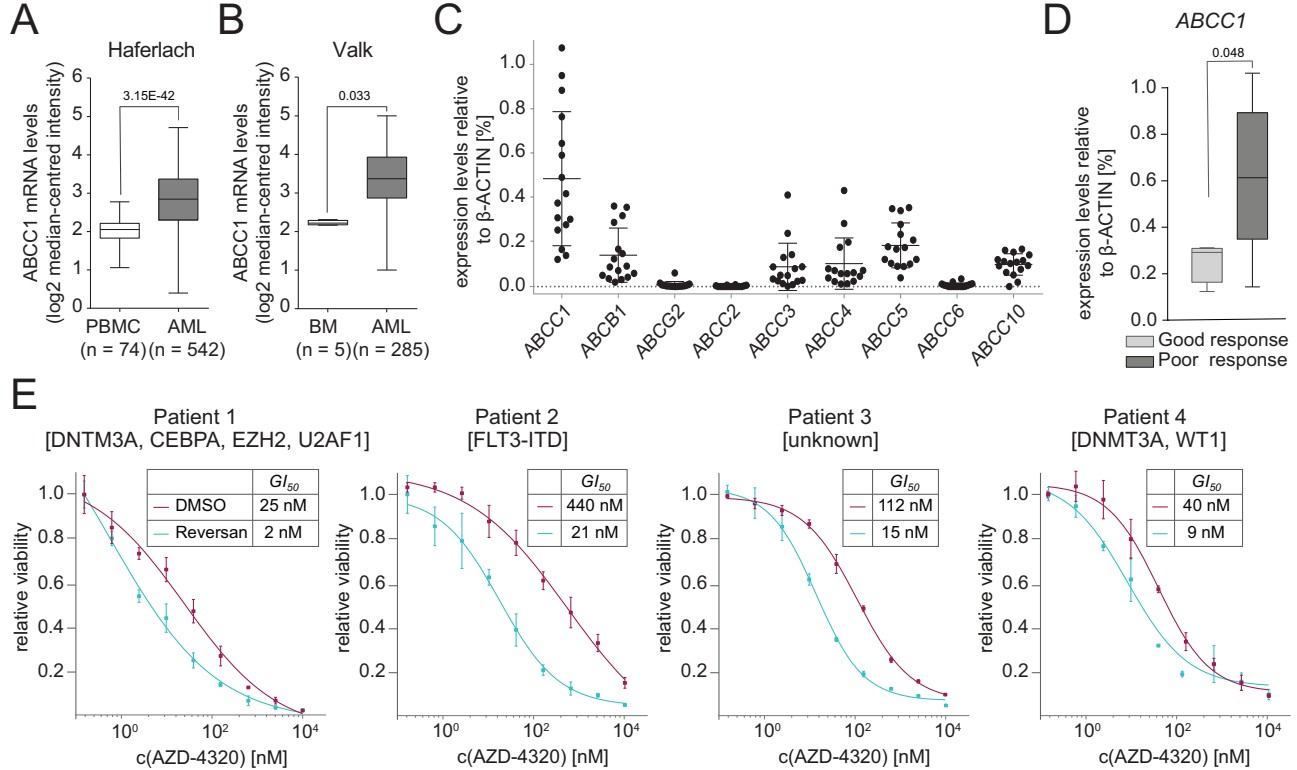

**Fig. 6 | ABCC1 levels dictate the response of AML to Venetoclax. A** *ABCC1* mRNA expression levels in AML patient samples ($n = 542$) compared to peripheral blood mononuclear cells (PBMCs) from healthy controls ($n = 74$). Data and statistical analysis were extracted from the Oncomine Platform from the Haferlach Leukemia dataset. Boxes represent interquartile ranges, horizontal lines represent the mean, whiskers indicate lower and upper limits. **B** *ABCC1* mRNA expression levels in AML patient samples ($n = 285$) compared to bone marrow ($n = 5$) from healthy controls. Data and statistical analysis were extracted from the Oncomine Platform from the Valk Leukemia dataset. Boxes represent interquartile ranges, horizontal lines represent the mean, whiskers indicate lower and upper limits. **C** Expression of ABC transporters in AML patient samples ($n = 16$ individual patient samples) relative to ACTB [%]. Analysis of expression levels of each patient sample was performed in duplicates. Data are presented as mean values ± SD. Detailed patient information is

listed in Supplementary Table 1. **D** Comparison of *ABCC1* expression levels in patients with good or poor response to Venetoclax treatment ($n = 14$ individual patient samples−same as in (**C**)−2 non-responders excluded). Boxes represent interquartile ranges; horizontal lines represent the median expression; whiskers indicate lower and upper limits of the respective patient cohort. Analysis of expression levels of each patient sample was performed in duplicates. Significance was determined with an unpaired Student's *t* test with two-tailed *P* value as indicated. **E** Dose−response curves of primary patient-derived AML cells co-treated with either DMSO or Reversan (patient 1 and 2, 5 µM, patient 3 and 4, 2.5 µM) and AZD-4320 for 3 days. Data are presented as mean values ± SD. $n = 2$ (patient 4)/$n = 3$ (patient 1–3) samples of the same patient. Gene names refer to the mutational status of the patients (Supplementary Table 1). **C**−**E** Source data are provided as a Source Data file.

results show that the effect of inhibiting GSH synthesis acts on top of increased intracellular drug concentrations that are caused by *ABCC1* loss and indicates that GSH is needed for efficient detoxification. Finally, combinatorial treatment with BSO potentiated the anti-proliferative effects of Venetoclax and AZD-4320 in patient-derived primary human AML cells (Fig. 7E). Like BSO, treatment with EA alone had negligible effect on AML cell survival (Supplementary Fig. 5E). However, combinatorial treatment with EA sensitized MOLM-13 and HL-60 cells to Venetoclax and AZD-4320 (Supplementary Fig. 5F). Co-treatment with either BSO or EA led to intracellular Venetoclax

accumulation in ABCC1-overexpressing HL-60 cells (Supplementary Fig. 5G), suggesting that interference with glutathione metabolism impairs Venetoclax efflux. In contrast, co-treatment with 5-Azacytidine, which has previously been shown to act synergistically with Venetoclax therapy by disrupting energy metabolism[56], had no impact on intracellular Venetoclax concentrations.

In summary, these data suggest that glutathione plays a vital role in the detoxification of BH3 mimetics and that perturbation of GSH metabolism increases the sensitivity of AML cells towards BH3 mimetics.

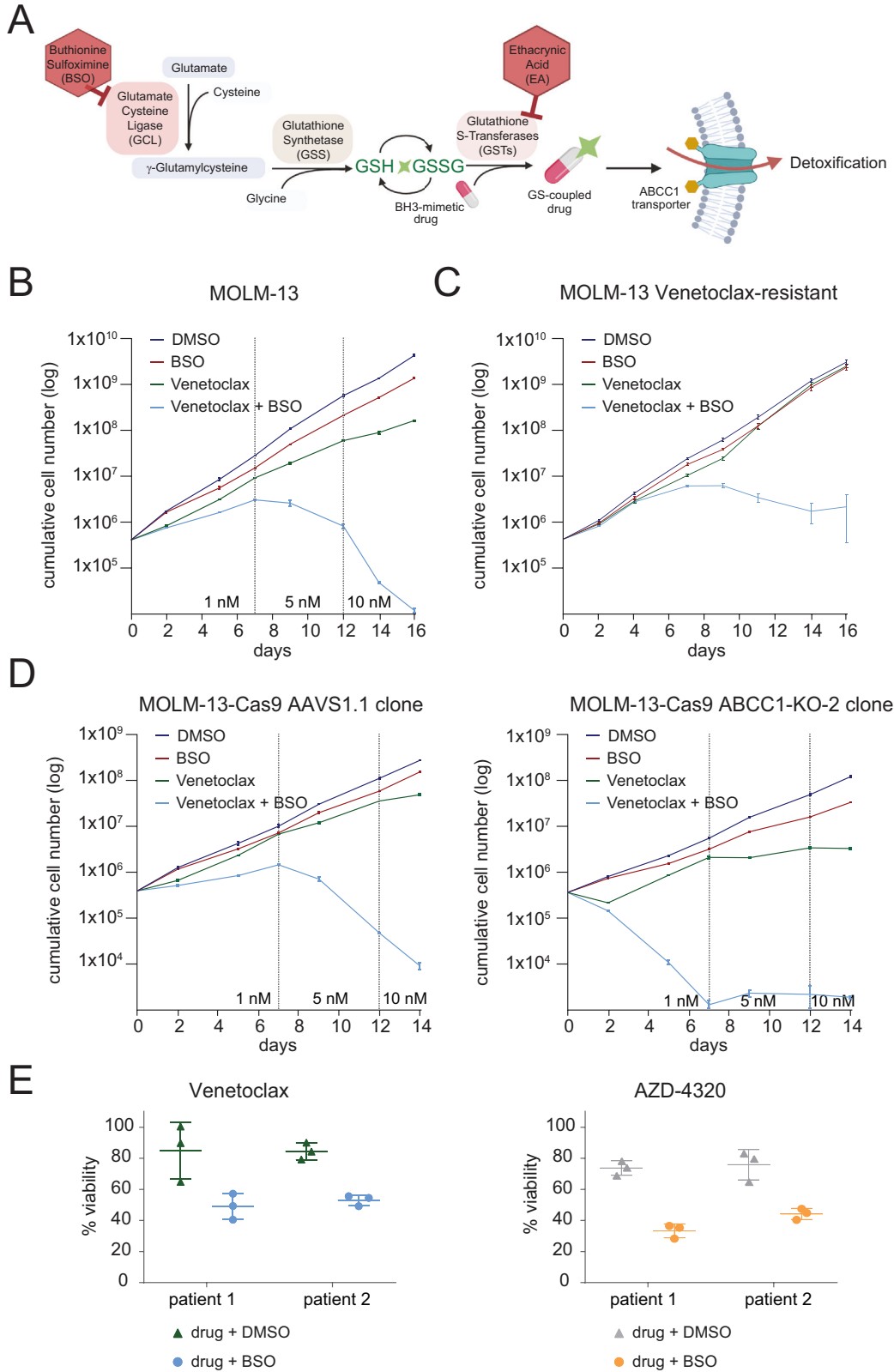

## Discussion

ABC transporters mediate the active transport of substances across cell membranes. They have protective roles in regulating the bioavailability, metabolism, distribution, and excretion of xenobiotics, detoxified drug conjugates and endogenous metabolites[25]. Using a targeted CRISPR/Cas9-based screen focused on the 48 genes encoding ABC transporters in the human genome we show that only four ABC

transporters are essential for AML cell growth. Among these genes, *ABCA3* was implicated in chemoresistance, especially in childhood AML[57,58]. Further, despite their association with chemoresistance[59], ABCE1 and ABCF1 are unable to transport substrates through membranes, as these molecules are devoid of a transmembrane domain. Finally, the TAP2 protein is involved in antigen presentation[60]. While the lethal effect of mutational inactivation of *ABCE1* is consistent with

**Fig. 7 | Glutathione metabolism modulates the sensitivity to BH3 mimetics.**
**A** Schematic depiction of glutathione metabolism in human cells. The enzyme glutamate-cysteine ligase (GCL) produces γ-glutamylcysteine from glutamate and cysteine. Then, glutathione synthetase (GSS) adds glycine to the C-terminus of γ-glutamylcysteine to form reduced glutathione (GSH), which can be oxidized to GSSG or conjugated to xenobiotic compounds by glutathione-S-transferases (GSTs). The synthesis of GSH can be inhibited using the GCL inhibitor Buthionine sulfoximine (BSO), while GSH-conjugation to drugs can be inhibited using the GSTs inhibitor Ethacrynic acid (EA). Figure was created using bioRender.com. **B** Growth curves of MOLM-13 cells treated with DMSO, 100 μM BSO, Venetoclax (increasing concentrations 1, 5 and 10 nM) or Venetoclax in combination with 100 μM BSO. Data are presented as mean values ± SD. $n = 2$ experimental replicates. **C** Growth curves of Venetoclax-resistant MOLM-13 cells treated with DMSO, 100 μM BSO, 1 μM Venetoclax or 1 μM Venetoclax in combination with 100 μM BSO. Data are presented as mean values ± SD. $n = 3$ experimental replicates. **D** Growth curves of MOLM-13-Cas9 knockout clones *AAVS1.1* (left), *ABCC1-KO-2* (right) treated with DMSO, 100 μM BSO, Venetoclax (increasing concentrations 1, 5 and 10 nM) or Venetoclax in combination with 100 μM BSO. Data are presented as mean values ± SD. $n = 2$ experimental replicates. **E** Relative viability of primary patient-derived AML cells treated with Venetoclax (left) or AZD-4320 (right) in combination with DMSO or 100 μM BSO for 3 days. Concentrations used were: patient 1: 9.8 nM Venetoclax and 2.4 nM AZD-4320; patient 2: 39 nM Venetoclax and 39 nM AZD-4320. Data were normalized to DMSO-treated controls and are presented as mean values ± SD. $n = 3$ experimental replicates. **B**–**E** Source data are provided as a Source Data file.

the literature[61], the roles of *ABCF1*, *ABCA3* and *TAP2* in AML cell growth are not well established and might reveal novel tissue-specific vulnerabilities.

In addition to their role in normal cellular physiology, ABC transporters can influence the pharmacokinetic parameters of drug absorption, distribution, metabolism, excretion, and toxicity[62–64]. ABC transporters play a crucial part in cancer drug resistance, as they can increase the cellular efflux of anticancer drugs[65]. However, it is incompletely understood which and how ABC transporters interact with targeted agents that have been added to the repertoire of cancer drugs recently. For instance, BCL-2 inhibitors mimic the binding of pro-apoptotic BH3-only proteins to BCL-2, thereby effectively inducing apoptosis[9]. The promising activity of the BCL-2 inhibitor Venetoclax in several combinatorial regimens has led to its approval for the treatment of AML patients[11]. While so far only selected ABC transporters, including ABCB1 and ABCG2, have been studied for their role in BCL-2 inhibitor transport[66,67], we investigated all 48 ABC transporters using an unbiased CRISPR/Cas9 screening approach in AML. We found that mutational disruption of *ABCC1* increased the sensitivity of AML cells to Venetoclax. This effect was not restricted to Venetoclax, as knock-out or pharmacologic inhibition of ABCC1 also sensitized AML cells to other BH3 mimetics, including ABT-737, Navitoclax and the novel BCL-2/BCL-xL inhibitor AZD-4320. These results indicate that conserved structural features of those molecules are responsible for the specificity of the effect. Strikingly, *ABCC1* inhibition was even able to re-sensitize resistant cells to Venetoclax treatment. In this case, drug resistance was not determined by differences in intracellular Venetoclax concentrations between sensitive and resistant cells, but due to increasing intracellular drug levels upon perturbation of ABCC1 expression. This is particularly relevant, as developing resistance is a major problem in Venetoclax therapy.

In contrast, inactivation of *ABCC1* had no impact on cell proliferation in multiple AML cell lines in the absence of drugs, which is in line with previous data[68], and ABCC1 inhibition did not sensitize AML cells to various other anticancer drugs that are used to treat AML, including the chemotherapeutic Cytarabine, FLT3-inibitors and IDH inhibitors. All these observations indicate that exquisitely specific and non-redundant cellular efflux mechanisms for small molecules exist that are different from the well-characterized functions of ABC transporters in the efflux of chemotherapeutic agents. The high specificity of the interaction between ABCC1 and BCL-2 inhibitors suggests that efflux transporters hold great potential for the development of rational cancer treatments that involve newly approved small molecule inhibitors.

A major problem associated with the use of BH3 mimetics is the lack of specific biomarkers that can guide their application. Several genetic lesions have been identified to predict the efficacy of Venetoclax, including mutations in *IDH1/2* and *SRSF2/ZRSR2*[69,70]. Recent studies proposed that high expression of *BCL2A1*, *CLEC7A* or *CD14* predicts Venetoclax sensitivity[71] and moreover, combinatorial levels of BCL-2-family members in AML leukemic stem cells determine the response to 5-Azacytidine in combination with Venetoclax treatment[72]. Different studies have linked ABC transporter expression levels to adverse prognosis in AML, in particular *ABCB1*, *ABCC1* and *ABCG2*[29,73]. Co-expression of multiple ABC transporters is generally associated with poor therapy response[30]. While these studies did not investigate the association of ABC transporter expression with Venetoclax therapy, we found that patients with poor response to Venetoclax displayed significantly elevated expression of ABCC1. Strikingly, the expression of other important ABC transporters did not differ between patient groups. Given that ABCC1 is among the highest expressed ABC transporters in AML and its selective role in efflux of BH3 mimetics, our results highlight the importance of future studies in larger patient cohorts to further establish the role of ABCC1 as a biomarker that predicts the response to BCL-2 inhibition in AML. More recently, an increasing body of evidence suggested that solid tumors might also be efficiently targeted by BCL-2 inhibitors[74–78]. Its broad expression pattern across various tissues[79] establishes ABCC1 as a strong candidate for a biomarker of drug response in these settings.

Our studies establish that glutathione metabolism modulates the efficacy of BCL-2 inhibitors. Inhibition of glutathione synthesis or glutathione conjugation sensitized AML cells to BCL-2 inhibition. In accordance, it was previously shown that BSO can re-sensitize *ABCC1*-overexpressing cells to Etoposide and Daunorubicin[80]. Together, these data suggest that the glutathione pathway could play a role in the detoxification of BH3 mimetics by GSH-drug conjugation. Although inhibition of ABCC1 potentiates the effect of BH3 mimetics, combinatorial treatment with ABCC1-inhibitors and BH3-mimetic drugs is currently not a therapeutic option for patients, as no ABCC1 inhibitor is approved for clinical use. The development of new inhibitors, usage of alternative delivery strategies or the use of antisense nucleotides could provide novel approaches for targeting ABC transporters. Alternatively, targeting of important co-factors, such as glutathione metabolism could be another promising strategy to increase drug efficacy. This is particularly relevant in the context of BH3 mimetics, as they display limited anticancer activity when used in monotherapy settings[81].

In conclusion, this work proposes ABCC1 as a functional biomarker for the response of AML cells to BH3 mimetics and suggests interference with ABCC1 function or GSH homeostasis as rational strategies to overcome Venetoclax resistance. Our data highlight that efflux pumps should be revisited as promising intervention points to enhance the effectiveness of small molecule inhibitors in leukemia and beyond.

## Methods
The research in this study complies with all relevant ethical regulations. Animal experiments were approved by the Institutional Ethics and Animal Welfare Committee and the National Authority according to §26ff. of Animal Experiments Act, Tierversuchsgesetz 2012 – TVS 2012 (license number BMWF 68.205/188-V/3b/2018, GZ-2021-0430018). Primary AML samples were obtained by bone marrow

puncture or venipuncture during routine investigations at the time of diagnosis. Cells were stored in a local biobank until used. All patients gave written informed consent before bone marrow or blood was obtained. The study was approved by the ethics committee of the Medical University of Vienna (EK-No: 1355/2021). Detailed information about the patients is listed in Supplementary Table 1.

## Cell culture

Stable *Cas9*-expressing clones of HL-60, MOLM-13 and THP-1 AML cells were used. All human AML cell lines, HL-60 (ACC 3), MOLM-13 (ACC 554), MV4-11(ACC 102), THP-1 (ACC 16), KG-1 (ACC 14), PL-21 (ACC 536) were purchased from DSMZ and validated using STR profiling. Human cell lines and murine MLL/AF9-NrasG12D-driven cells (RN2)[35] were cultivated in RPMI-1640 medium (Gibco, USA) supplemented with 10% fetal bovine serum (Thermo Fisher Scientific, USA), 100 U/ml penicillin, 100 μg/ml streptomycin, 4 mM L-glutamine, 50 μM 2-mercaptoethanol (all Gibco, USA) and 20 mM HEPES (Sigma-Aldrich, USA) at 37 °C, 5% $CO_2$, and 95% humidity. Mycoplasma contamination was routinely tested using the Venor GeM Classic Mycoplasma Detection Kit (Lonza, Basel, Switzerland). Lenti-X 293T (Takara, Cat. Nr.: 632180) and Platinum-E cells (Cell Biolabs, Cat. Nr.: RV-101) were cultivated in DMEM (Gibco, USA) supplemented with 10% fetal bovine serum (Thermo Fisher Scientific, USA), 100 U/ml penicillin, 100 μg/ml streptomycin (Gibco, USA) and 4 mM L-Glutamine (Gibco, USA). Primary patient-derived AML cells were isolated using Ficoll gradients and were cultured in IMDM (Thermo Fisher Scientific, USA) supplemented with 15% BIT 9500 Serum Substitute (STEMCELL Technologies, Canada), 100 ng/ml stem cell factor (SCF) (ImmunoTools, Germany), 50 ng/ml FLT3L (ImmunoTools, Germany), 20 ng/ml interleukin-3 (IL-3) (ImmunoTools, Germany), 20 ng/ml G-CSF (ImmunoTools, Germany), 0.1 mM 2-mercaptoethanol (Gibco, USA), 50 μg/ml gentamicin (Thermo Fisher Scientific, USA), 10 μg/ml ciprofloxin (Thermo Fisher Scientific, USA), 500 nM SR1 (APExBIO, USA) and 1 μM UM279 (APExBIO, USA) at 37 °C, 5% $CO_2$, and 95% humidity. Venetoclax-resistant MOLM-13 cells were generated by treatment with increasing Venetoclax concentrations for up to 2.5 months until they exhibited stable growth kinetics in medium supplemented with 1 μM Venetoclax. Once cells were Venetoclax-resistant, *Cas9* was introduced, and the pool of Cas9-expressing cells was used for subsequent experiments.

## Plasmids and constructs

Single guide RNAs (sgRNAs, Supplementary Table 2) were designed using the VBC score sgRNA prediction tool[82] and cloned into a lentiviral vector enabling expression of the sgRNA and IRFP670 (pLenti-hU6-sgRNA-PGK-IRFP670). The ABCC1 overexpression construct was a gift from Scott Dixon[83]. For the reconstitution experiment, silent point mutations were introduced into the ABCC1 cDNA using PCR mutagenesis to avoid targeting by *ABCC1* sgRNAs and the product was cloned into a pMSCV-PGK-BlastR retroviral expression vector using Gateway cloning. The psPAX2 (Addgene plasmid #12260) and pMD2.G (Addgene plasmid #12259) were gifts from Didier Trono. pCMV-gag/pol was acquired from Cell Biolabs, San Diego, USA.

## Viral transduction

For retroviral transduction of target cells, Platinum-E cells were co-transfected with the transfer vector and pCMV-gag/pol using polyethylene imine (Polysciences, USA). For lentiviral transduction, Lenti-X-293T cells were transfected with transfer vector, psPAX2 and pMD2.G using polyethylene imine (Polysciences, USA). Virus was harvested 48 h and 72 h post transfection, filtered (0.45 μm) and supplemented with polybrene (4 μg/ml, Merck, Germany). Target cells were spinoculated in biological triplicates by addition of viral supernatants (1:10 diluted) and centrifugation for 90 min at 900×*g* at room temperature.

## Compounds

Venetoclax, AZD-4320, Navitoclax, ABT-737, AZD-5991, 5-Azacytidine (5-Aza), Etoposide and MK-571 were purchased from MedChemExpress (USA). Midostaurin, Gilteritinib, Ivosidenib and Enasidenib were acquired from Selleck Chemicals (USA). Reversan was obtained from Merck (Germany). Ethacrynic acid (EA) was purchased from Sigma-Aldrich (USA). All compounds were diluted in DMSO. Buthionine sulfoximine (BSO) was purchased from Merck (Germany) and diluted in ddH₂O. For in vivo experiments, AZD-4320 was dissolved in 30% 2-Hydroxypropyl-β-cyclodextrin (HP-β-CD, Merck (Germany) solution in ddH₂O (3 mM stock).

## Competitive cell proliferation assays

To assess the effect of CRISPR/*Cas9*-mediated gene targeting on cell proliferation, *Cas9*-expressing cells were transduced with sgRNA/IRFP670 expression vectors and IRFP670-positive cells were monitored at regular intervals by flow cytometry using an IntelliCyt IQue-Screener Plus (BioScience, Sartorius Group, Germany). The corresponding gating strategy for IRFP670+ cells is depicted in Supplementary Fig. 6A. Values were normalized to day 4 post lentiviral transduction. In drug incubation experiments, values were normalized to the first day of treatment and to DMSO control. The area under the curve (AUC) after 30 days of treatment of DMSO-treated MOLM-13-*Cas9* cells expressing sgRNAs targeting ABC transporters is plotted against the AUC of drug treated cells. If the AUC (DMSO; 0.95) > AUC (Drug; 0.8) we speak of synergistic effects; if the AUC (DMSO; 1.0) < AUC (Drug; 1.1) we speak of antagonistic effects (Supplementary Fig. 1C).

## Growth curves

Cells were seeded in triplicates and treated with either DMSO or with the respective compounds at the indicated concentrations for 48 or 72 h. Cells were counted at regular intervals. Growth rates and cumulative cell numbers were calculated using Microsoft Excel and the Prism 6.0.1 (GraphPad, USA) software. Growth rate was determined according to Eq. (1).

$$\frac{Cell\ number\ t(1)}{Cell\ number\ t(0)} * splitting\ factor \qquad (1)$$

## Cell viability assays/dose−response curves

Cells were seeded in 96-well plates in triplicates and treated with the indicated drugs in a dilution series. To determine synergistic drug effects, a matrix of dilution series of two drugs at indicated concentrations was established. Combinatorial responses were calculated based on the BLISS reference model[84] using SynergyFinder[85] and are displayed as the relative deviation from BLISS additivity. For determining combinatorial treatment effects with either Reversan, MK-571, BSO or Ethacrynic acid, the medium was supplemented with the respective compound 24 h prior to assay preparation to impair transporter/enzyme function before the addition of the second drug. Cell viability was determined using the CellTiter-Glo® Luminescent Cell Viability Assay (Promega, USA), on a Spark multimode microplate reader (TECAN, Switzerland) after 5 days. In experiments with primary patient-derived cells, responses were measured after 3 days of incubation. Dose−response curves were calculated using the Prism 6.0.1 (GraphPad, USA) software.

## Genotyping PCR analysis

Genomic DNA was extracted using the Quick gDNA Miniprep Kit (ZYMO Research, USA) followed by PCR amplification of the targeted region in the *ABCC1* gene using LA Taq DNA Polymerase (Takara Bio, Japan) according to standard laboratory protocols. PCR products were purified and analyzed by Sanger Sequencing.

Tracking of Indels by Decomposition (TIDE)[86] was used to identify the type of insertions and/or deletions in 15 clones. To identify the exact mutations in *ABCC1* knockout clones, PCR products were subcloned using the TA Cloning® Kit (Thermo Fisher Scientific, USA) and two clones were analyzed by Sanger Sequencing. Primers are listed in Supplementary Table 2.

## Xenotransplantation-based models and drug treatment

NOD.Cg-Prkdcscid Il2rgtm1Wjl Tg(CMV-IL-3,CSF2,KITLG)1Eav/MloySzJ mice expressing human IL-3, GM-CSF and SCF on a NSG background (NSG-S mice) were purchased from Jackson Laboratory (Bar Harbor, ME, USA). NSG-S mice were kept in specific opportunistic pathogen free quality (SOPF) under stringently controlled standard conditions in individually ventilated cages and were fed with SSNIFF Haltungsfutter (CHOW standard, 10 mm pellets; Catalog-No. V1534-000) ad libitum. In total, $1 \times 10^6$ cells (clones MOLM-13 *AAVS1.1* and MOLM-13 *ABCC1-KO-2*) were injected into the tail vein of the NSG-S mice. Engraftment and disease progression was monitored by whole-body fluorescence imaging (IRFP670 expression) using the IVIS optical imaging system (PerkinElmer, USA). Signal quantification was performed using the Living Image analysis software (PerkinElmer, USA) with standardized circular regions of interests covering the mouse trunk and extremities, including blank correction of signal strength. Two days after transplantation, mice were treated with AZD-4320 or vehicle (10 mg/kg, once a week) by intravenous injection into the tail vein. All recipient mice were between 16-20 weeks old and both male and female mice were represented in each cohort. Mice developing first disease symptoms, including lack of grooming, reduced agility and/or hind limb motility, were sacrificed and IRFP670-positive cells in the bone marrow were quantified on a FACS Canto II flow cytometer (BD Biosciences), and analyzed with the FlowJo software (FlowJo, LLC). The flow cytometric gating strategy for IRFP670+ cells is depicted in Supplementary Fig. 6B.

## Real-time PCR analysis

Total RNA was extracted using the Qiagen RNeasy Mini Kit (Qiagen, Germany). Reverse transcription was done using the RevertAid first-strand cDNA synthesis Kit (Thermo Fisher Scientific, USA) and quantitative PCR was performed using the SsoAdvanced Universal SYBR Green Supermix (Bio-Rad, USA) on a Bio-Rad CFX96-Real-Time PCR Detection System. Relative expression levels were determined by normalizing C(t) values to human β-Actin (ACTB) using the 2-ΔΔC(t) method. Absolute gene expression was determined by calculating the percentage of gene expression relative to human β-actin using the Eq. (2)[87]. Primer sequences are listed in Supplementary Table 2.

$$2^{(-\Delta C(t))} * 100 \qquad (2)$$

## Flow cytometry−intracellular staining

Cell samples were washed and stained with Zombie Aqua™ viability dye (BioLegend, USA), followed by fixation in 2% ROTI®Histofix (Carl Roth, Germany) for 15 min. For cell permeabilization, samples were washed, re-suspended in 0.2%Triton X-100 (PanReac AppliChem, USA) in PBS + 10% FCS (FACS buffer) and incubated for 15 min. After incubation in 0.1% Triton-FACS buffer for 30 min, permeabilized cells were incubated with a primary antibody against ABCC1 (anti-MRP1, ab24102, lot: GR284248-19, Abcam, UK; dilution 1:200) followed by a secondary fluorescence-labelled antibody (Goat anti Mouse IgG, FITC, Cat-No: P31582, lot: 1915874, Thermo Fisher Scientific, Austria; dilution 1:200). Stained samples were measured using a FACS Canto II flow cytometer (BD Biosciences), and analyzed with the FlowJo software (FlowJo, LLC). The flow cytometric gating strategy is depicted in Supplementary Fig. 6C.

## Measurement of intracellular drug concentrations

Cells were treated with 500 nM AZD-4320 for 2.5 h before pelleting. In time course experiments, HL-60 ABCC1 OE cells were treated with 1 μM Venetoclax or AZD-4320 for 2 h, washed twice with ice-cold PBS and incubated in drug free media for 30 or 60 min before pelleting and harvesting of supernatants. Co-treatment experiments were performed by incubating cells with the respective compounds (Reversan, BSO, Ethacrynic Acid, 5-Azacytidine) for 2 h, followed by addition of Venetoclax. After another 2 h incubation, cells were washed twice with ice-cold PBS and re-suspended in Venetoclax-free media for 2 h before harvesting pellets. Pelleted cells were washed twice with PBS and extracted using a solvent mixture of methanol:acetonitrile:H$_2$O (2:2:1) (Merck, Austria, ACS grade pure chemicals). Lysis was performed in 3 cycles of vortexing for 30 s, incubation in liquid nitrogen for 1 min followed by subsequent thawing at room temperature and sonication for 10 min in an ice-cold water bath. For protein precipitation, samples were incubated for 1 h at −20 °C, followed by centrifugation at 10,000×*g* for 15 min at 4 °C. Supernatants were shock frozen and stored at −80 °C. Non-targeted LC-MS/MS (liquid chromatography-tandem mass spectrometry) analysis was performed with an Ultimate 3000 HPLC system (Thermo Fisher Scientific) coupled to a Q Exactive Focus mass spectrometer (Thermo Fisher Scientific). In total, 1 μL of each extract was injected to the respective column, employing a flow rate of 100 μL/min for separation. For reversed phase chromatography, an ACQUITY UPLC HSS T3 column (150 mm × 2.1 mm; 1.8 μm) with VanGuard precolumn (Waters Corporation) was used. A 20 min gradient of 99% A (0.1% formic acid in water) to 60% B (acetonitrile with 0.1% formic acid) was used. For HILIC (hydrophilic interaction chromatography), an iHILIC®-(P) Classic, PEEK column, (100 mm × 2.1 mm, 5 μm) with a guard column (HILICON) was used. Here, a 26 min gradient from 90% A (acetonitrile) to 80% B (25 mM ammonia bicarbonate in water) was used. For both separations, mass spectra were acquired with a mass resolution of 70,000 in polarity switching mode with full MS acquisition from *m/z* 70 to *m/z* 1045. Fragmentation data were collected with data-dependent acquisition in a pooled sample (normalized collision energy: 25%). Data analysis and statistical evaluation was performed with Compound Discoverer 3.0 (Thermo Fisher Scientific) searching public databases and our in-house spectral library.

For quantification of Venetoclax and AZD-4320 metabolite extracts were analyzed by triple quadrupole LC-MS/MS. In brief, 1 μl of each sample was injected onto a Kinetex (Phenomenex) C18 column (100 Å, 150 × 2.1 mm) connected to the respective guard column. The HPLC (Vanquish UHPLC system, Thermo Fisher Scientific) was directly coupled via electrospray ionization to a TSQ Altis mass spectrometer (Thermo Fisher Scientific). A 4-min-long linear gradient from 98% A (1% acetonitrile, 0.1% formic acid in water) to 90% B (0.1% formic acid in acetonitrile) at a flow rate of 100 μl/min has been used, employing a column temperature of 30 °C. The selected reaction monitoring (SRM) mode was used for detection and external quantification, using the following transitions in the positive ion mode: *m/z* 434.2 to *m/z* 321.2 (Venetoclax, qualifier), *m/z* 868.3 to *m/z* 321.2 (Venetoclax, quantifier), *m/z* 945.3 to *m/z* 404.2 (AZD-4320), *m/z* 150.1 to *m/z* 133.1 (methionine), *m/z* 148.1 to *m/z* 84.1 (glutamic acid) and *m/z* 120.1 to *m/z* 74.1 (threonine). Retention times and optimal collision energies have been defined by using authentic standards. Data interpretation was performed using TraceFinder (Thermo Fisher Scientific). For targeted analyses of drug concentrations, ion counts of Venetoclax or AZD-4320 were normalized to intracellular levels of Threonine to control for potential differences in cell numbers.

## Calcein-acetoxymethyl ester (Calcein-AM) transporter assay

Cells were seeded at $5 \times 10^5$ cells/ml in the respective cell culture medium and treated with indicated concentrations of drugs or DMSO for 45 h. After 5 min of incubation at room temperature in plain

RPMI-1640 medium, 100 nM Calcein-AM (Sigma-Aldrich, Germany) was added, and cells were incubated for 10 min at 37 °C. The reaction was stopped by addition of access ice-cold plain medium. Cells were harvested and re-suspended in PBS. The median fluorescence intensity (MFI) of Calcein, the fluorescent product in the Calcein-AM assay of live cells was recorded in the FITC channel of a FACS Canto II flow cytometer (BD Biosciences), and analyzed with the FlowJo software (FlowJo, LLC). The flow cytometric gating strategy is depicted in Supplementary Fig. 6D.

### Data analysis and statistics and reproducibility
The Prism 6.0.1 software (GraphPad, USA) was used for statistical analyses, and data are shown as mean ± SD. Experiments were performed in duplicates/triplicates and repeated at least three times. The individual sample size is reported in figure legends. The unpaired Student's $t$ test was used for $P$ value determination. Results were considered significant when $P < 0.05$. Exact $P$ values are indicated in each figure. No statistical method was used to predefine sample size. The experiments were not randomized. The investigators were not blinded to allocation during experiments and outcome assessment.

### Reporting summary
Further information on research design is available in the Nature Portfolio Reporting Summary linked to this article.

## Data availability
All data generated in this study are provided in the Source Data file. Publicly available datasets used are not in the Source Data file. Gene expression data for *ABCC1* were extracted from the Haferlach Leukemia, Valk Leukemia or TCGA Leukemia datasets (reporter: 202804_at) using the Oncomine™ Research Premium Edition database (Thermo Fisher, USA)[88], accessed in July, 2021. Gene expression analysis of ABC transporters in AML patients in the BeatAML dataset[47] was accessed in January, 2021 from the NCI Genomic Data Commons: https://gdc.cancer.gov/about-data/publications/BEATAML1-0-COHORT-2018. The Ordino database was used to extract gene expression data in human AML cell lines (KG-1, THP-1, PL-21, MV4-11, HL-60, MOLM-13). The Ordino database contains data from: The Cancer Genome Atlas (TCGA), the Cancer Cell Line Encyclopedia (CCLE), two depletion screen datasets[89,90]; data extracted in August, 2021. Source data are provided with this paper.

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

## Acknowledgements

We would like to thank all members of the Grebien laboratory for stimulating discussions. We thank Elizabeth Heyes and Thomas Eder for help with data analysis, Anna Falkner and Edwin Rzepa for help with cloning and assay preparations, and Gabriele Stefanzl and Daniela Berger for collecting and sampling of primary AML cells. We are grateful to the Dixon group (http://www.dixonlaboratory.com/) for kindly sharing constructs for ABCC1 cDNA expression. We acknowledge Gerhard Ecker for discussions about substrate specificity of ABC transporters. We thank the animal caretakers at the University of Veterinary Medicine Vienna for expert animal handling. LC-MS/MS analysis was performed by the Metabolomics Facility at Vienna BioCenter Core Facilities (VBCF), which is a member of the Vienna BioCenter (VBC) and funded by the City of Vienna through the Vienna Business Agency. Research in the Grebien lab is supported by the Austrian Science Fund (projects P-35628, P35298 and TAI-490) and by the European Research Council (ERC) under the European Union's Horizon 2020 research and innovation program (grant agreement no. 636855 to F.G.). The project also received funding from NCI grant no. R21 CA26740 and funds from the Endowed Haas Chair in Genetics to M.A. J.E. was supported by grant no. 857935 from the Austrian Research Promotion Agency (FFG). J.S. is a recipient of a DOC Fellowship of the Austrian Academy of Sciences at the University of Veterinary Medicine and the Ludwig Boltzmann Institute for Cancer Research, respectively. P.V. was supported by the Austrian Science Fund (FWF) SFB grant F4704-B20. Research at the IMP is generously supported by Boehringer Ingelheim.

## Author contributions

J.E., J.S. and F.G. conceptualized the study. J.E., J.S., M.P., G.M., S.T., B.Z.C., H.N., R.M., G.S., J.Z., T.K., M.A., W.R.S., P.V. and F.G. performed the investigations. J.E., J.S. and F.G. wrote the manuscript with input from all co-authors.

## Competing interests

J.Z. is a founder, shareholder and scientific advisor of Quantro Therapeutics GmbH. J.Z. and the Zuber laboratory receive research support and funding from Boehringer Ingelheim. The remaining authors declare no competing interests.
