## [Peer Review File · Nature Communications]

This manuscript has been previously reviewed at another journal that is not operating a transparent peer review scheme. This document only contains reviewer comments and rebuttal letters for versions considered at *Nature Communications*.

REVIEWER COMMENTS

Reviewer #3 (Remarks to the Author):

In their revised manuscript, the authors added novel information that in large part addresses my previous concerns. The authors now show that perturbation of GSH signaling by BSO and EA causes an increase of intracellular Vtx drug levels, and they correctly state in the manuscript that it is likely that GST detoxification may play a role to modulate Vtx resistance that is in addition to its effects on ABCC1-dependent drug efflux.

The remaining specific questions are:

1. The authors rendered MOLM13 and MV4-11 cells Vtx-resistant by long-term exposure to the drug. They show that both, KO and pharmacological inhibition (Reversan) of ABCC1 resensitize the resistant cell lines to Vtx. Notably, Reversan re-established sensitivity towards a Vtx concentration of 1nM, whereas for the KO, only data at Vtx concentrations of 500nM are shown for MOLM13 (Fig. S3I). As presented, the data do not allow to estimate the level to which ABCC1 KO and Reversan treatment re-establish sensitivity towards Vtx. Please, provide these data and discuss, if large differences of the effects of Reversan versus ABCC1 KO were to be observed.

2. Along the same lines: the authors now nicely show that intracellular Vtx concentrations are higher after ABCC1 KO in THP1 (Fig. 5D) cells, and in HL60 cells after Reversan treatment (Fig. S3F). They also show that Vtx-resistant MOLM13 cells express similar ABCC1 levels as the parentals (Fig. S3G). They claim that Vtx resistance is at least partially driven by ABCC1 activity. How do the intracellular Vtx (or AZD4320) concentrations for a given extracellular concentration compare between the MOLM13 and/or MV4-11 resistant and sensitive cell lines? Ideally, intracellular concentrations should be shown as a dose-response function for the sensitive versus the resistant cell lines.

Minor:

1. The "slight", but significant difference in ABCC1 expression between de novo and sAML patients may well be worth noting in the manuscript, given the higher number of clinical resistance of sAML patients versus de novo patients.
2. Please, correctly label the y-axis in the synergy plots.

RESPONSE TO REVIEWERS' COMMENTS

We would like to thank all reviewers for their efforts, feedback and helpful comments. We are very glad to see that all of them appear to endorse the study. In the following section, we would like to clarify the open points that were raised by Reviewer #3.

Resulting changes to the text are highlighted in yellow in the manuscript file.

REVIEWER COMMENTS

Reviewer #3 (Remarks to the Author):

In their revised manuscript, the authors added novel information that in large part addresses my previous concerns. The authors now show that perturbation of GSH signaling by BSO and EA causes an increase of intracellular Vtx drug levels, and they correctly state in the manuscript that it is likely that GST detoxification may play a role to modulate Vtx resistance that is in addition to its effects on ABCC1-dependent drug efflux.

The remaining specific questions are:

1. The authors rendered MOLM13 and MV4-11 cells Vtx-resistant by long-term exposure to the drug. They show that both, KO and pharmacological inhibition (Reversan) of ABCC1 resensitize the resistant cell lines to Vtx. Notably, Reversan re-established sensitivity towards a Vtx concentration of 1nM, whereas for the KO, only data at Vtx concentrations of 500nM are shown for MOLM13 (Fig. S3I). As presented, the data do not allow to estimate the level to which ABCC1 KO and Reversan treatment re-establish sensitivity towards Vtx. Please, provide these data and discuss, if large differences of the effects of Reversan versus ABCC1 KO were to be observed.

We thank the reviewer for pointing this out. To address this point, we inactivated *ABCC1* in Venetoclax-resistant MOLM-13 cells using the CRISPR/Cas9 technology. Disruption of the *ABCC1* gene was validated by genotyping/TIDE analysis (Brinkman et al., 2014) in pooled cell populations. We measured the Venetoclax sensitivity of *ABCC1*-mutant vs *ABCC1*-wildtype Venetoclax-resistant MOLM-13 cells in a short-term dose response experiment. In parallel, we determined the effect of Reversan-mediated ABCC1 inhibition in re-sensitizing Venetoclax-resistant MOLM-13 cells to Venetoclax using the same assay. As shown below (**Reviewer Figure 1**) both genetic and pharmacological interference with ABCC1 increased Venetoclax sensitivity in Venetoclax-resistant cells.

Interestingly, however, we found that in this short 5-day drug sensitivity assay, CRISPR/Cas9-mediated targeting of *ABCC1* caused a re-sensitization of resistant cells to 1.4-fold lower Venetoclax concentrations, while Reversan-mediated ABCC1 inhibition sensitized cells to 5-fold lower Venetoclax concentrations. Several factors could be responsible for this difference in drug sensitivity. First, CRISPR/Cas9-mediated targeting is often causing incomplete loss of the target protein, particularly when pooled cell populations are used (as in this case). Thus, remaining low levels of ABCC1 protein might mask a stronger effect on Venetoclax sensitivity in short-term drug treatment assays. In line with this, low doses of Venetoclax were sufficient to re-sensitize resistant cells upon CRISPR/Cas9-mediated *ABCC1* targeting in long-term competitive proliferation assays (**Fig. S3I**). These issues could be circumvented by selecting

single cell-derived clones that feature complete ABCC1 loss and/or performing long-term drug treatment assays. But as these experiments involve time consuming and laborious processes, we decided to use the CRISPR/Cas9-targeted cell pool in short-term drug treatment experiments to address the reviewer's question as quickly as possible. In contrast, pharmacological ABCC1 inhibition using Reversan results in an instant inhibitory effect, which translates into lower GI₅₀ values in conventional drug sensitivity assays. However, the exact target spectrum of Reversan is not known, and it might be possible that the drug has inhibitory activity towards other ABC transporters in addition to ABCC1. Therefore, both approaches have several shortcomings that need to be considered when they are directly compared.

Nevertheless, our data clearly show that both genetic and pharmacological perturbation of ABCC1 can re-sensitize resistant cells to Venetoclax. However, the levels and extent of sensitization are likely dependent on the experimental setup and the chosen read-out. We argue that a thorough comparison of all aspects of the effects of genetic vs. pharmacological ABCC1 perturbation might require much more time and is therefore beyond the scope of this revision. We hope that the reviewer agrees with this but still values our initial efforts to address this point.

Reviewer Figure 1. Genetic and pharmacological ABCC1 interference increased Venetoclax sensitivity in MOLM13 Venetoclax-resistant cells

Heatmap of GI₅₀ values [nM] of 5-day viability assays with indicated sgRNAs and drugs in MOLM-13 Venetoclax-resistant cells (n=3).

2. Along the same lines: the authors now nicely show that intracellular Vtx concentrations are higher after ABCC1 KO in THP1 (Fig. 5D) cells, and in HL60 cells after Reversan treatment (Fig. S3F). They also show that Vtx-resistant MOLM13 cells express similar ABCC1 levels as the parentals (Fig. S3G). They claim that Vtx resistance is at least partially driven by ABCC1 activity. How do the intracellular Vtx (or AZD4320) concentrations for a given extracellular concentration compare between the MOLM13 and/or MV4-11 resistant and sensitive cell lines? Ideally, intracellular concentrations should be shown as a dose-response function for the sensitive versus the resistant cell lines.

We investigated intracellular Venetoclax concentrations in HL-60 cells (Venetoclax-sensitive) vs THP-1 cells (intrinsically Venetoclax-resistant). As shown below (**Reviewer Figure 2**) both cell lines displayed comparable intracellular Venetoclax concentrations upon treatment. Moreover, as the reviewer also pointed out, we found that MOLM-13 Venetoclax-resistant cells express similar ABCC1 levels as MOLM-13 WT cells (Fig.S3G). These datasets indicate that intracellular Venetoclax concentrations are comparable between Venetoclax-sensitive and Venetoclax-resistant cells. We therefore favor a scenario in which drug resistance is not determined by differences in intracellular Venetoclax concentrations, but knockout or

pharmacological interference with ABCC1 can effectively re-sensitize Venetoclax resistant cells due to increasing intracellular drug levels. We added a statement addressing this point to the discussion section of the revised manuscript.

Reviewer Figure 2. Intracellular Venetoclax concentrations in HL-60 vs THP-1 cells

Ion count ratios of intracellular Venetoclax to Threonine (control amino acid) of HL-60 (Venetoclax-sensitive) or THP-1 (Venetoclax-resistant) cells upon treatment with Venetoclax (1 μ M) as determined by LC-MS/MS (n=2).

Minor:

1. The “slight”, but significant difference in ABCC1 expression between de novo and sAML patients may well be worth noting in the manuscript, given the higher number of clinical resistance of sAML patients versus de novo patients.

We agree with the reviewer and have added the plot showing the difference in ABCC1 levels between de novo AML and sAML to **Supplementary Figure 4D** of the revised manuscript.

2. Please, correctly label the y-axis in the synergy plots.

We thank the reviewer for spotting this, we have fixed this issue in the manuscript.

References

Brinkman, E.K., Chen, T., Amendola, M., and Van Steensel, B. (2014). Easy quantitative assessment of genome editing by sequence trace decomposition. *Nucleic Acids Res.* 42, e168.

REVIEWERS' COMMENTS

Reviewer #3 (Remarks to the Author):

The authors show that (1) in a short term assay KO and pharmacological inhibition of ABCC1 have similar effects of different magnitudes and argue, that both interventions have their shortcomings, which renders them difficult to directly compare; (2) they argue that it is rather the change of IC drug concentrations than the absolute concentrations that determine the response of the cells.

While one could still argue that thorough analyses may have increased the strength of their points, I agree with the authors that concern number (1) is answered by putting the responses into a similar magnitude of \pm 3-fold, and that they (2) provide enough IC concentration data to make their point that this contributed to ABCC1-mediated Vtx resistance.

RESPONSE TO REVIEWERS' COMMENTS

We would like to thank all reviewers for their efforts, feedback and helpful comments. We are very glad to see that all of them appear to endorse the study.

REVIEWER COMMENTS

Reviewer #3 (Remarks to the Author):

The authors show that (1) in a short term assay KO and pharmacological inhibition of ABCC1 have similar effects of different magnitudes and argue, that both interventions have their shortcomings, which renders them difficult to directly compare; (2) they argue that it is rather the change of IC drug concentrations than the absolute concentrations that determine the response of the cells.

While one could still argue that thorough analyses may have increased the strength of their points, I agree with the authors that concern number (1) is answered by putting the responses into a similar magnitude of ?? 3-fold, and that they (2) provide enough IC concentration data to make their point that this contributed to ABCC1-mediated Vtx resistance.

We thank the reviewer for this remark. We agree with the reviewer that deeper analyses on this topic and other assays determining resistance mechanisms will be valuable in future studies, but would have been beyond the scope of our manuscript. Nevertheless, we are happy to see that the reviewer agrees that our IC concentration data are robust.